# Pediatric Infective Endocarditis: A Literature Review

**DOI:** 10.3390/jcm11113217

**Published:** 2022-06-05

**Authors:** Lourdes Vicent, Raquel Luna, Manuel Martínez-Sellés

**Affiliations:** 1Servicio de Cardiología, Hospital Universitario 12 de Octubre, and Instituto de Investigación Sanitaria Hospital 12 de Octubre (imas12), 28041 Madrid, Spain; mlourdesvicent@gmail.com (L.V.); raquel.lunalop@gmail.com (R.L.); 2Servicio de Cardiología, Hospital Universitario Gregorio Marañón, Calle Doctor Esquerdo, 46, 28007 Madrid, Spain; 3Faculty of Medicine, Universidad Complutense de Madrid, 28040 Madrid, Spain; 4Faculty of Medicine, Universidad Europea de Madrid, 28670 Madrid, Spain

**Keywords:** infective endocarditis, pediatrics, children

## Abstract

Infective endocarditis in children is a rare entity that poses multiple challenges. A history of congenital heart disease is the most common risk factor, although in recent years, other emerging predisposing conditions have gained relevance, such as central venous catheters carriers or children with chronic debilitating conditions; cases in previously healthy children with no medical history are also seen. Diagnosis is complex, although it has improved with the use of multimodal imaging techniques. Antibiotic treatment should be started early, according to causative microorganism and risk factors. Complications are frequent and continue to cause significant morbidity. Most studies have been conducted in adults and have been generalized to the pediatric population, with subsequent limitations. Our manuscript presents a comprehensive review of pediatric infective endocarditis, including recent advances in diagnosis and management.

## 1. Epidemiology

Infective endocarditis (IE) is a rare condition, that is even more uncommon in children. The incidence of pediatric IE has been estimated to be 0.43–0.69 cases per 100,000 children–year [1,2]. This low incidence makes it difficult to obtain evidence regarding the best diagnostic and therapeutic approach of IE at those ages. To try to overcome this issue, studies frequently use large retrospective databases of medical records based on the International Classification of Diseases. However, those studies are biased on coding errors and scarce clinical data [3]. Pediatric IE is thought to have distinctive characteristics in terms of presentation and underlying predisposing factors compared to adults. This is a narrative review, including the most recent evidence and scientific advances in pediatric IE.

## 2. Predisposing Factors

In children with IE, congenital heart disease has been described as the main predisposing factor [1,4,5]. The correct identification of risk factors for IE is essential to identify populations that benefit from antibiotic prophylaxis. The main predisposing factors of pediatric IE are depicted in Table 1.

Globally, the frequency of IE in children is lower than in adults, but in recent decades, an increase has been observed due to the rise in risk factors, such as congenital heart disease or intravascular devices (central catheters, pacing leads, etc.). The predisposing factors are similar to those of the adult population, however, the vulnerability to some of these factors may be greater in the pediatric population. In fact, a younger age is a predisposing factor for IE in cases of bacteremia, and younger patients are at greater risk of worse outcomes [6].

### 2.1. Congenital Heart Disease

About 50–70% of pediatric IE is seen in patients with congenital heart disease [7,8,9], which is the main predisposing condition. Conversely, in adult patients, the most common heart diseases predisposing to IE include degenerative valve disease or valve replacement surgery with prosthesis implantation, which are significantly more common than in children [10]. The type of congenital heart disease influences IE risk. Cyanotic and complex congenital heart disease, left-sided defects, or endocardial cushion defects are the main variants associated with higher risk [5,11,12,13]. Patients with congenital heart disease at 18 years of age have a risk of IE that is superimposable to patients of 81 years of age without congenital heart disease, which represents a risk 75 times greater than that of the general population [14]. Recent reconstructive cardiac surgery (<6 months) in patients with congenital heart disease is an important predisposing factor for IE [5]. There is also an increased risk in patients with a shunt or residual heart defects after corrective surgery, those who receive prosthetic valve implantation (either surgical or transcatheter), or in patients with prosthetic materials.

Congenital heart disease in adults with IE is uncommon, but its prevalence is increasing, including complex congenital heart disease and patients with previous corrective surgeries, who have a high risk of IE, so an increase in the frequency of IE can be expected in this group [13,15].

### 2.2. Acquired Risk Factors

In adults, degenerative valve disease and intravenous drug abuse are relevant risk factors for IE, although they are rare in children [16]. In adults without previous heart disease, chronic kidney disease on hemodialysis is a common risk factor for IE [17]. Children with serious underlying chronic conditions have also an increased risk of IE. Some examples are those with immunodeficiency, patients with frequent invasive interventions or with central catheters (chemotherapy, nutrition, or haemodialysis), and institutionalized patients [8]. Children with cancer present several concurrent risk factors, such as tumor-related immunosuppression, nutritional deficit, chemotherapy, and the placement of catheters and central vascular accesses [7,18,19]. The frequency of IE in this group has increased substantially in recent years and the rise in cases will probably continue in the next coming years as the survival of pediatric cancer keeps improving. About 10–25% of pediatric IE is already seen in children with serious illnesses or acquired conditions [8,20].

The place of acquisition is also relevant, since children with hospital-acquired IE are sicker [21]. Those with hospital-acquired IE are usually younger and commonly have risk factors such as diabetes, hemodialysis, malignancies, are carriers of central venous catheters, and often have had a recent surgery in the previous 8 weeks [21]. In addition, the microbiological etiology is different in nosocomial IE, with a higher frequency of *candida* [22], *Gram-negative*, or *coagulase-negative staphylococcus* infections [21].

Rheumatic fever, which in the past was the main worldwide predisposing factor of IE, has almost disappeared in developed countries, but continues to be a public health problem in developing countries [23,24]. Therefore, although it is uncommon, it should be kept in mind, especially in patients from certain geographical areas.

Oral health is an important factor in both children and adults. Episodes of transient bacteremia have been observed with dental procedures and with daily tooth brushing [25,26]. However, periodontal disease and gingivitis are more common in adults.

### 2.3. No Known Risk Factors

In a minority of cases, IE occurs in children with no known chronic conditions or congenital heart disease [27]. These previously healthy children are usually older compared to those with predisposing conditions [27]. Conversely, adults with no known risk factors tend to be younger [17]. In some cases, IE occurrence has been related to dental procedures or extensive skin lacerations [8,27]. It is unclear if IE in these healthy individuals could be attributable to a subclinical and previously unapparent heart condition. Another possibility would be the presence of unknown systemic risk factors.

Previous healthy children that develop IE have a worse prognosis with an increased risk of complications [8] compared to those with an identifiable predisposing condition. This may be related to infections caused by more aggressive pathogen [8]. Conversely, in adults without previous heart disease, the prognosis is more favorable [17].

## 3. Pathophysiology

IE is caused by an infection of the cardiac endothelium [28]. The pathophysiology is shared in adult and pediatric patients with IE. As endothelium is resistant to microbiological colonization, most patients present added disruptive factors [28]:(1)Predisposing heart disease generating turbulent blood flow that produces endocardial damage that causes a thrombotic and fibrotic reaction;(2)Bloodstream bacterial or *fungal* infection (mainly from mucosal or skin source);(3)Foreign materials (catheters, heart conditions repaired with prosthetic material, cardiac implantable electronic devices) that facilitate the deposit of fibrin, platelets, and thrombus formation.

These conditions might precipitate the appearance of an infected mass (or “vegetation”) stuck to the endothelium of cardiac valves or the foreign materials. The turbulent blood flow caused by structural lesions in congenital heart disease might produce endocardial damage that could cause, first, a thrombotic and fibrotic reaction aimed at repairing tissue damage (first event). Such jet-induced thrombosis with platelet deposition and fibrin is the first of four concatenated events causing IE by creating a minimal non-bacterial thrombotic vegetation. In the case of bacteremia (second event), an adherence of micro-organisms to the non-bacterial thrombotic vegetation can take place, with the subsequent deposition of more fibrin and platelets that cover the infective agents (third event) favoring their multiplication (fourth event) [29,30,31].

## 4. Clinical Presentation

The clinical presentation of IE in children is highly conditioned by age [20,28]. In adolescents, the symptoms and findings are quite similar to the ones seen in adult patients: fever, loss of appetite, and malaise as evidence of a systemic infectious process. A history of congenital heart disease or previous cardiac defects that may or may not have required surgical correction may increase suspicion. A new murmur or a change in a previous murmur may be heard.

In cases of subacute onset, it is common to find progressive deterioration in functional class, with persistent fatigue, malaise, or growth retardation. Patients with IE may present evidence of acute heart failure due to valve destruction.

Regarding systemic manifestations, there may be renal immunological involvement (glomerulonephritis), seizures, or cerebral infarction due to cerebral embolisms. Immunological phenomena (Osler nodes, Janeway spots [Figure 1]) are less common in children than in adults.

In newborn children, the manifestations of IE are nonspecific and can be confused with other more common systemic infections, such as pneumonia or urinary tract infections. Tachycardia, feeding refusal, vomiting, or high fever are common. Seizures may appear as evidence of central nervous system involvement. Skin or joint manifestations are less usual [20,32,33]. In fact, in most cases, the younger the age, the more challenging the diagnosis.

Risk factors for IE (especially the presence of congenital heart disease) and location vary according to the patient’s medical history [10]. In children with a history of congenital heart disease, the right location is common, especially in the pulmonary valve [10,34]. The presence of previous corrective surgeries with the use of prosthetic material, grafts, patches, or conduits pose an increased risk of infection and a diagnostic challenge, since the visualization of these conduits is complex with the usual imaging techniques, such as echocardiography [10,33] (Figure 2).

## 5. Microbiology

Causal pathogens vary according to the underlying conditions. In addition, there have been changes in IE epidemiology in recent decades. Globally, *S. aureus* remains the dominant causative pathogen both in children and adults [35]. *Streptococci* infections, primarily the viridans group, are also very frequent [35]. Specifically, in the case of children, *Staphylococcus species* are the most common (especially in children without heart disease), followed by *Streptococcus species* [1,10]. However, for children with underlying heart disease, the viridans group *Streptococcus* is the most common cause [1,10] (Figure 3). As in adults, IE from *Gram-negative* organisms is rare in children [1,36,37].

Blood cultures are one of the fundamental pillars for the diagnosis of IE, together with the physical examination and echocardiography findings. An adequate extraction and processing of the sample with an incubation period of at least 5 days is necessary to guarantee the adequate reliability of the blood culture in the diagnosis of IE [1,20,38]. However, 5 to 7% of patients with IE have negative blood cultures [6,39,40]. In these cases, previous treatment with antibiotics, infections by slow-growing germs, or inadequate microbiological techniques may explain the absence of the identification of the pathogen [16,20,37]. In a very low proportion of cases, it could be a non-infectious endocarditis, in the context of autoimmune processes such as lupus, although this situation is very rare in children. With advances in microbiological diagnostic techniques, including molecular techniques (polymerase chain reaction) and serology, the percentage of IE cases with an unknown microbiological cause has been reduced [39].

Table 2 shows the main microbiological etiology of IE in children, according to the presence of heart disease in a large cohort of patients with IE [10].

## 6. Diagnosis

IE is diagnosed based on the modified Duke criteria [41,42]. The diagnosis of IE is complex in children because clinical manifestations are frequently nonspecific and may be confused with more common conditions. A high level of suspicion is required. The Duke criteria (Table 3) combine clinical, echocardiographic, and microbiological findings. The Duke criteria were primarily tested in adult patients and few studies have evaluated the diagnostic performance of the current criteria for the diagnosis of pediatric IE [6,42,43]. The modified Duke classification is more sensitive in diagnosing IE in children [44], but still, 12% failed to be classified as “definite” [42]. Definite IE is defined as two major OR one major + three minor criteria, whereas possible IE requires one major + one minor OR three minor criteria [41,45,46]. The modified Duke criteria include the major criteria (blood culture positive for a typical microorganism (i.e., *Staphylococcus aureus*, *Enterococcus*, *Streptococci viridans*, and echocardiography findings such as valvular vegetation)) and the minor criteria (predisposing condition, fever, embolic phenomena, immunologic phenomena: glomerulonephritis, Osler’s nodes, Roth’s spots, and rheumatoid factor) [41,45].

### 6.1. Echocardiography and Imaging Techniques in the Diagnosis of IE in Children

The incorporation of echocardiography in IE diagnostic criteria has notably increased sensitivity. There are three major findings that are considered major criteria for IE diagnosis [47]: (a) the presence of vegetations [Figure 4]; (b) abscesses; (c) a new dehiscence of a valvular prosthesis. Other abnormal findings are considered as minor criteria [47].

Transthoracic echocardiography is the first imaging technique to perform, since it is highly available, provides relevant information, and is useful for follow-up and complication assessments. In cases of high diagnostic suspicion with a negative transthoracic echocardiography, it is recommended to extend the study with a transesophageal echocardiogram [46]. Although transesophageal echocardiography in adults has shown a better resolution, and therefore, a greater sensitivity, in children, both tests have comparable diagnostic performances, at least in those with a body weight < 60 kg [32]. Therefore, we can deduce that, as a first approximation, evaluation with a transthoracic echocardiography would be sufficient, especially in the group of younger or lighter children, due to their better acoustic window [32].

Transesophageal echocardiography has several technical considerations [48]:(1)it requires frequently general anesthesia or sedatives;(2)a need to adapt the tube size, especially in younger children;(3)a possible vascular or airway compromise, especially with inappropriate probe sizes;(4)the interpretation of images may be complex and requires specific training.

Despite these considerations, transesophageal echocardiography is useful in children with congenital heart disease, especially those who have had previous surgeries or children who are planning to undergo surgical treatment for IE [48]. Intra operative monitoring with transesophageal echocardiography is usually needed for guiding cardiac surgery. Transthoracic echocardiography has advantages, but its diagnostic yield decreases when the probability of IE is low and in children with a structurally normal heart and negative microbiological tests [49,50].

Another important limitation of echocardiography, especially transthoracic echocardiography, is that the proportion of false negatives could reach 70% in patients with complex congenital heart disease [51], due to the difficulty in differentiating the vegetations from other prominent cardiac structures, or surgical remains, as well as the poor visualization of shunts or conduits [51].

Intracardiac echocardiography is a promising tool in the diagnosis of IE. As we have previously mentioned, conventional transthoracic and transesophageal echocardiography may have a poor diagnostic performance in children with cardiac devices or with previous corrective surgeries. In these cases, intracardiac echocardiography, which is a very versatile technique, can overcome these limitations [52,53,54].

### 6.2. Multimodality Cardiovascular Imaging

Advanced cardiovascular imaging modalities such as cardiac magnetic resonance imaging, computed tomography angiography, or positron emission tomography in combination with computed tomography (PET-CT) have gained interest in recent years, due to the lower sensitivity of echocardiography for the diagnosis of IE in children with corrected congenital heart disease, especially those with mechanical prosthetic valves, conduits, stents, or ventricular assist devices [55,56,57]. High-resolution multislice gated cardiac CT has a high added value for diagnosing paravalvular complications such as abscess, but there is scarce evidence for the diagnosis of infective endocarditis in children [34]. Moreover, PET-CT can be a valuable diagnostic tool for children with a fever of an unknown origin [58] and this indication has been validated in children.

In a recently published case, PET-CT was an essential diagnostic test, allowing the reclassification of cases of “possible” or “rejected” IE as “definite” IE, which allows an earlier diagnosis, and thus, specific management [59]. Given the characteristics of pediatric patients with IE, which are different from those of adults, these multimodal diagnostic imaging techniques are of great importance, as more than half of the children with IE have a history of congenital heart disease [10,11,46]. In these cases, the most frequent location of IE is the right heart valves, which are poorly visualized on echocardiography, especially when a previous surgical correction has been performed with the implantation of prostheses or ducts [59,60].

These multimodal imaging techniques are very versatile, since they allow a differential diagnosis with other infective processes, confirm the diagnosis in cases with inconclusive echocardiographic techniques, and therefore, have a great impact on subsequent antimicrobial therapy [61]. The sensitivity is close to 100% and its reliability has also been demonstrated in children [56,57,61]. PET-CT is very useful in detecting septic emboli and also in areas with little clinical expression such as the lungs or the spleen [55]. Despite its high diagnostic reliability, PET-CT has a number of limitations that must be taken into account [62]:exposure to ionizing radiation, which have a greater negative impact in children;interference with physiological activity (i.e., central nervous system, reactive intestinal ganglia, etc.);it requires preparation with a specific diet the days before the test;it requires differential diagnosis with other situations of tissue hypermetabolism, for example, after surgery or neoplasms. Therefore, the reliability of PET-CT for the diagnosis of IE in patients with recent cardiac surgery may be limited.

### 6.3. Blood Cultures

It is of great importance to guarantee sterility at the time of sample collection. The volume of blood to be extracted varies according to the age and weight of the patient, and can range from 1 mL in patients < 5 kg under 1 year of age, up to 10–15 mL in patients of 15 years of age [63]. The sensitivity of blood cultures increases with each additional mL of blood drawn, but this can be difficult to achieve in young patients with a low body surface area [64]. The causes of persistently positive blood cultures should also be assessed, that is, those with a repeated isolation of the same microorganism.

Blood cultures should be interpreted according to the following principles [63]:Blood cultures should be taken as soon as possible, ideally < 1 h;Three sets of blood cultures should be drawn, in aerobic and anaerobic culture bottles. If a limited volume of blood is available, aerobic culture media should be preferred, as anaerobic IE is very rare;Usually, the incubation time necessary to observe bacterial growth and obtain antibiotic sensitivity is 24–48 h;If there is a high clinical suspicion of bacterial IE and negative blood cultures, it is important to consider molecular techniques to increase sensitivity, such as PCR, to detect bacterial DNA or the 16S subunit of bacterial ribosomes, which is species-specific. Bacterial identification by PCR also allows an earlier result to be obtained [63];It is of great importance to take blood cultures in the absence of antibiotic treatment, in those stable patients in whom the clinical situation allows the antibiotic to be discontinued;Consider repeating the cultures at 48–72 h in the presence of persistent or recurrent symptoms (fever, inflammatory parameters in laboratory tests, systemic inflammatory response).

## 7. Differential Diagnosis

IE diagnosis can be challenging. Table 4 shows the more common differential diagnoses to consider among children with fever and clinical suspicion of IE.

## 8. Complications

Complications can be classified as cardiac and extracardiac. Complications are more frequent in children without known heart diseases, especially those <2 years of age [21]. Furthermore, complications of IE in children are less common in children than in adults [6].

Cardiac complications are the most common, and can occur in up to 50% of patients [65,66]. These complications are caused by the local destructive effect of the infection on the valvular apparatus and surrounding cardiac structures [67]. Heart failure is the most common cause of mortality in patients with IE [9], and is caused by valvular insufficiency and regurgitation [66]. In children with congenital heart disease corrected with a stent, conduit, or valve implantation, infection and vegetation formation can cause the dysfunction of these structures due to obstruction by vegetations or thrombi [51,68]. Heart failure is more frequent in left-sided native valve infective endocarditis [67]. This complication occurs less often in patients with congenital heart disease, since right valves are more commonly affected, and the hemodynamic profile and prognosis are more favorable in this group [10,69,70]. As heart failure is an indication for cardiac surgery, it requires prompt diagnosis and management, as it is an important predictor of poor prognosis [65,67].

Other cardiac complications include the annular extension of infection with a paravalvular abscess that may cause conduction disturbances, sinus of Valsalva perforation, suppurative pericarditis, or intracardiac fistula [46]. Risk is higher in patients with aortic valve IE.

Extracardiac complications are caused by embolic phenomena of the cardiac vegetations, sepsis and a systemic inflammatory response, or immune-mediated mechanisms (i.e., glomerulonephritis, vasculitis). Metastatic infection may manifest as embolization, abscess, or mycotic aneurysms. Embolic phenomena may be present in 10–40% of the patients [71]. Mycotic aneurysms are rare in children, but it is worth mentioning the cases that appear in children with coarctation of the aorta, since there are cases reported in the literature and mortality is close to 100% if surgery is not performed [72,73,74].

A study that analyzed neurological complications specifically in children with IE found that they could occur in up to a quarter of cases, with ischemic stroke being the most common [75]. Intracranial hemorrhage can appear in children with mycotic cerebral aneurysms, or due to a hemorrhagic transformation of a previous ischemic stroke, or in anticoagulated patients [75,76]. Other nonspecific manifestations include seizures, meningitis, or brain abscess.

In recent years, the importance of brain imaging techniques has been pointed out to assess neurological damage in patients with IE before surgery, even in asymptomatic cases, since it has been described that only half of patients with ischemic neurological lesions have symptoms [77,78,79]. The results of neurological examinations such as magnetic resonance imaging of the brain are important to guide the timing of surgery, since with the presence of intracranial bleeding or ischemic stroke it is recommended that surgery is delayed for between 2 and 4 weeks, although in these cases, the decision can be complex and other factors must be taken into consideration, such as the risk of recurrent embolism, sepsis, or heart failure [80].

Renal failure has a multifactorial origin, since there are various potential causative mechanisms including antibiotic toxicity, hemodynamic instability due to sepsis or heart failure, renal embolisms, or immunological phenomena [46]. Glomerulonephritis is an infrequent complication that requires, in addition to antibiotic treatment directed at IE, the administration of immunosuppressive drugs, including the possibility of plasmapheresis [81,82]. Urinalysis may show hematuria, proteinuria, and red cell casts that are suggestive of glomerulonephritis.

Table 5 shows risk factors associated with a higher probability of complications.

## 9. Treatment

### 9.1. Antimicrobial Therapy

The antibiotic regimen used will depend on the type of microorganism, the susceptibility of the antibiogram, and the patient’s risk factors, including underlying heart disease [20,46]. Regarding the duration of antibiotic treatment, in general, it should be extended in patients with prosthetic valves or corrective surgeries with the implantation of prosthetic materials [20,46]. The route of administration for antibiotic treatment will be intravenous in children, avoiding intramuscular administration due to the risk of complications and the lower muscle mass. Antibiotics with bactericidal action are preferred over bacteriostatic.

Outpatient treatment could be considered in children without complications or high-risk factors, with negative control blood cultures, who remain afebrile, and in whom it is possible to guarantee therapeutic adherence, the collaboration of parents, and close supervision by cooperative parents and a home health nurse [83,84].

In seriously ill patients in whom an antibiogram is not available, or in cases with negative blood cultures, empirical antibiotic therapy will be started as soon as possible (Table 6).

### 9.2. Surgery Timing and Indications

The indications for surgery in the treatment of IE (Table 7) in children are similar to those in adults. The approach must be multidisciplinary, since the indication is often related to the appearance of a complication, so the time to indicate surgery can sometimes be complex (i.e., recent stroke) [70]. In this assessment, both the indication and the timing of cardiac surgery are of equal importance. In general, the surgery will be performed in the days following the indication, during hospitalization. The delay in intervention is associated with a greater probability of the extension and dissemination of the infection, and embolisms. It is desirable that the patient receives antibiotic treatment, with several days of antibiotics before cardiac surgery being performed, in order to achieve a better infection control [70].

With regard to neurological complications, it is reasonable to perform brain imaging techniques (brain CT or magnetic resonance imaging) in patients with left-side IE in order to detect possible stroke or intracranial bleeding, also in asymptomatic patients [70].

The new concept of vegetation size adjusted for the patient’s body surface area has been related to the risk of mortality [85]. This variable is of great interest, given that, to date, most of the recommendations in children have been extrapolated from the guidelines and studies carried out in adults. Due to the smaller size, vegetation of a few millimeters may have a high risk of complications in children.

The type of surgical intervention (repair vs. replacement) will be conditioned by the extent of valve damage due to IE. It may be necessary to use materials such as grafts to cover locally destructive defects such as abscesses, fistulas, or valve leaflet perforations. If possible, it is desirable to avoid valve replacement in actively growing children, due to the possibility of a future prosthetic mismatch [86]. Reinterventions are often needed, especially in patients with small prosthesis [86].

### 9.3. Other Therapeutic Considerations

Nutrition: Malnutrition is a poor prognostic factor that leads to increased morbidity, the prolonged length of hospital stays, and poorer surgical outcomes [87]. The negative impact of malnutrition on immunity translates into a greater occurrence of complications such as embolisms, heart failure, or sepsis. It is especially important to prevent and treat this problem early, with specifically targeted interventions that should include patients with home hospitalization.

Exercise and cardiac rehabilitation: IE may be associated with severe sequelae that may be caused by hospitalization deconditioning, cardiac surgery, or neurological complications (central nervous system embolisms, intracranial hemorrhages, etc.). The benefit of cardiac rehabilitation for adults with IE may be controversial, as one study did not show an effect on mental health or physical capacity [88]. In other clinical contexts, the benefit of cardiac rehabilitation is relevant in children with severe congenital heart disease, improving their physical capacity and psychological well-being [89,90], so it seems reasonable to also refer children with IE to these rehabilitation programs as soon as possible, when they are clinically stable.

## 10. Prognosis

Mortality in children with IE is still high, and the sequelae are also serious. Globally, mortality is around 1–5%, although it is higher in patients with IE complications [1,36,37,91,92]. In adults, mortality is even higher, about 10% [93]. Specifically, infection by *S. aureus* is associated with a very high mortality and more than half of the children have complications, including admission to the intensive care unit [94,95]. The presence of congenital heart disease in itself does not imply a worse prognosis in patients with IE [96]. On the other hand, other factors, such as preexisting comorbidities or the male gender, were associated with a higher risk of complications and mortality [96,97].

The proportion of patients who present a recurrence of IE is high, approximately 5% [98,99]. Risk factors for recurrences are related to host factors, antibiotic treatment, and surgical outcome. The persistence of predisposing factors, such as poor oral hygiene, nosocomial acquisition, prosthetic valve IE, renal or liver disease, or being a carrier of central catheters or intracardiac devices, significantly increases the probability of a relapse [99]. Excessively short or incomplete antibiotic regimen or surgery with incomplete resection of the affected tissue are other risk factors for a relapse of IE [99,100]. Patients with new episodes of IE present a worse evolution, with higher mortality and complications. Therefore, children with the aforementioned risk factors are a group that should be closely monitored.

## 11. Prophylaxis

IE prophylaxis has changed according to more recent recommendations, restricting the indication to only patients at the highest risk undergoing procedures likely to result in bacteremia with a microorganism that has the potential ability to cause IE [46,101] (Table 8). Despite this more restrictive indication, no clear increase in the incidence of IE hospitalizations has been found after 2007 [102,103,104]. High-risk procedures, for which IE antibiotic prophylaxis would be recommended, include oral procedures with the manipulation of the gingival or periapical dental mucosa or perforation of the oral mucosa [46,101]. Other respiratory, gastrointestinal or urinary procedures are excluded from this recommendation. The antibiotic used for prophylaxis will be oral *amoxicillin*, and in allergic patients, *clindamycin*, 30–60 min before the intervention [46,101,105].

It is desirable that parents and children with IE have careful oral hygiene. In addition, it is advisable to avoid procedures that involve a disruption of the skin barrier, such as piercings or tattoos, especially in very high-risk patients.

## 12. Conclusions

IE in children is still a challenging condition. Despite advances, a high suspicion is needed for diagnosis, and mortality and morbidity continue to be high. Children with IE are a group that presents a high rate of events during follow-up, including the need for follow-up reintervention or relapses of IE. Congenital heart disease continues to be the most common risk factor, but other factors associated with nosocomial IE are growing in frequency. Studies are needed to generalize the outpatient and home-based treatment in patients who meet criteria for a low risk of complications.

## Figures and Tables

**Figure 1 jcm-11-03217-f001:**
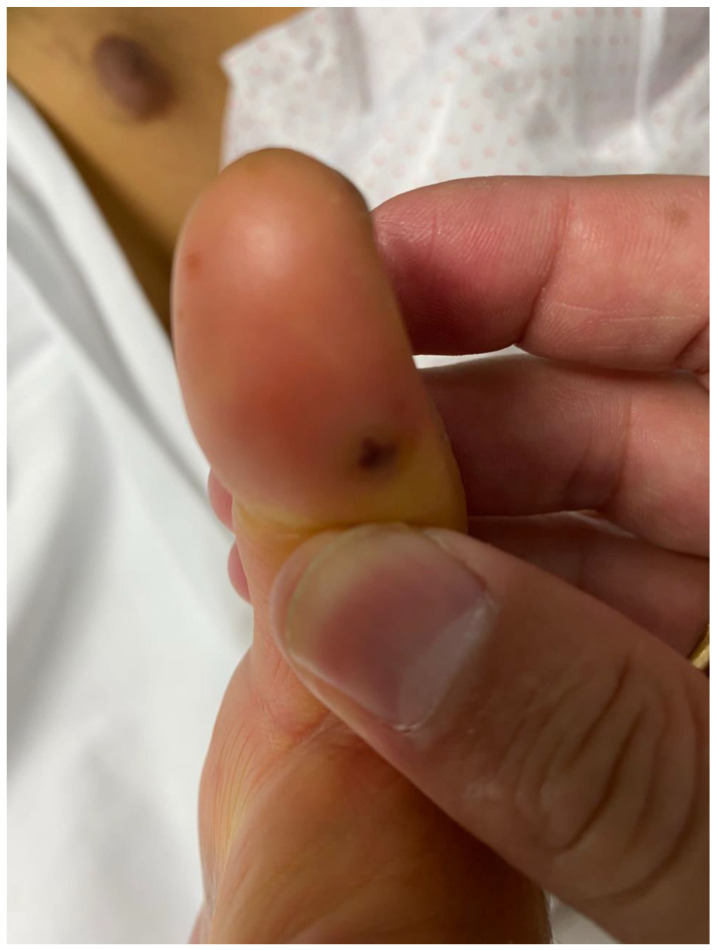
Janeway lesions in a 15-year-old adolescent with aortic valve infective endocarditis: hemorrhagic macules of the palms and soles that are due to septic emboli.

**Figure 2 jcm-11-03217-f002:**
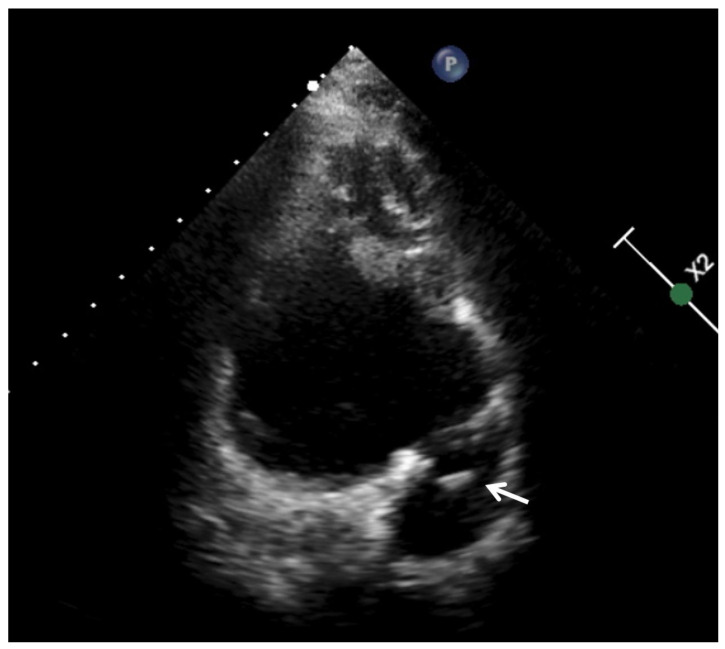
A 16-year-old woman with tetralogy of Fallot and pulmonary atresia. Contegra conduit endocarditis by *Streptococcus* sanguis.

**Figure 3 jcm-11-03217-f003:**
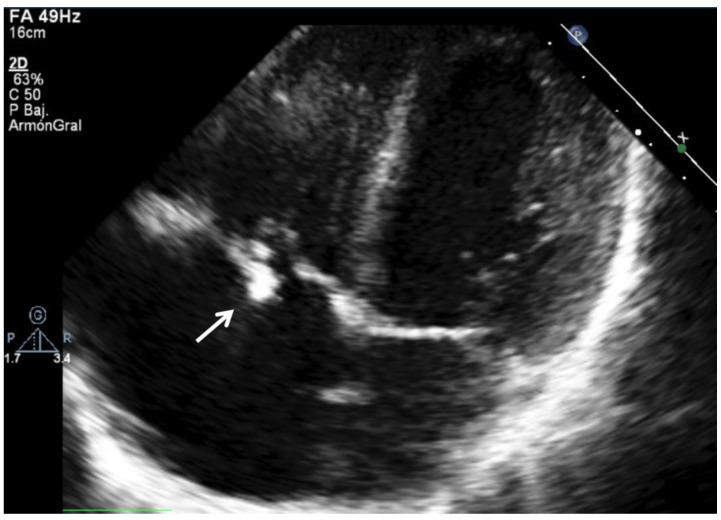
A 16-year-old male with native tricuspid valve infective endocarditis due to Viridans *Streptococcus*. Previous history of CHD with restrictive perimembranous ventricular septal defect, secondary moderate–severe tricuspid regurgitation, and septic pulmonary embolisms.

**Figure 4 jcm-11-03217-f004:**
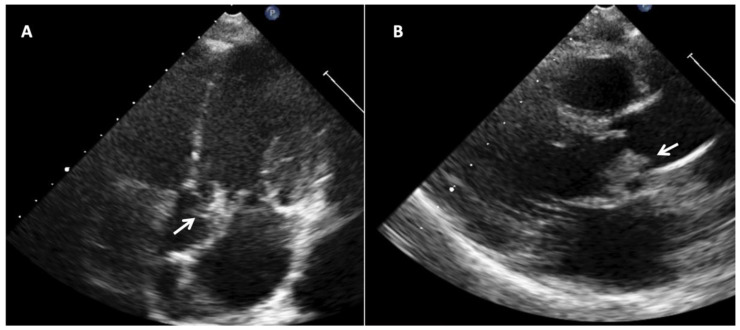
Infective endocarditis in a 13-year-old boy with bicuspid aortic valve due to Aggregatibacter aphrophilus (HACEK group). (**A**) Apical 5-chamber view showing aortic valve vegetation. (**B**) Paraesternal long axis with aortic valve showing aortic valve thickening and vegetation.

**Table 1 jcm-11-03217-t001:** Predisposing factors of infective endocarditis in children.

Congenital Heart Disease	Vulnerable: Acquired Risk Factors	Previous Healthy
Cyanotic disease	Immunodeficiency	Dental procedures
Recent cardiac surgery Left-sided lesions Endocardial cushion defects Recent corrective cardiac surgery	Cancer Hemodialysis	Skin infections/lacerations

**Table 2 jcm-11-03217-t002:** Etiology of pediatric infective endocarditis according to the history of congenital heart disease (CHD) [10].

No CHD	CHD
*S. aureus* 35–40%	*S. aureus* 25–30%
*Streptococcus* spp. 35–40%	*Coagulase-negative Staphylococci* 10–15%
*Enterococcus* 11%	*Streptococcus* spp. 20–30%
*Gram-negative* bacilli 10–15%	*Enterococcus* 3%
*Coagulase-negative Staphylococci* 0–5%	*Gram-negative bacilli* 25–30%

**Table 3 jcm-11-03217-t003:** Modified Duke criteria for diagnosis of infective endocarditis.

Infective Endocarditis–Modified Duke Criteria *
Major criteria
Blood culture positive for typical microorganism (i.e., *Staphylococcus aureus*, *Enterococcus*, *Streptococci viridans*)
Echocardiogram showing valvular vegetation
Minor criteria
Predisposing cardiac condition or injection drug use
Temperature > 38 °C
Embolic phenomena
Immunologic phenomena (glomerulonephritis, Osler’s nodes, Roth’s spots, and rheumatoid factor)
Positive blood culture not meeting above criteria

* Definite IE: 2 major OR 1 major + 3 minor criteria. Possible IE: 1 major + 1 minor OR 3 minor criteria.

**Table 4 jcm-11-03217-t004:** Differential diagnoses of infective endocarditis in children.

Infective Endocarditis Differential Diagnosis in Children
Familial Mediterranean Fever, juvenile rheumatoid arthritis
Rheumatic fever
Acute myocarditis
Pneumonia
Kawasaki disease
Acute myelocytic leukemia
Bacterial meningitis
Childhood vasculitis
Rheumatic diseases
Infections complicated with septicemia (i.e., soft tissues, urinary, etc.)

**Table 5 jcm-11-03217-t005:** Predisposing factors for complications in children with infective endocarditis.

Risk Factors for IE Complications
Size of the vegetation > 1 cm
Younger age, prematurity
*Staphyloccocus aureus*, *fungal* infection
No known heart disease
Left-sided valvular lesion
Complex cyanotic congenital heart disease
Higher white blood cell counts and plasma C-reactive protein
Persistent fever

**Table 6 jcm-11-03217-t006:** Antimicrobial regimens according to etiology in pediatric infective endocarditis (IE) with the recommended week (w) duration.

Causative Microorganism	Antibiotic Regimen	
Unknown agent	*Ampicillin* + *gentamicin* + *oxacillin* *Vancomycin* + *gentamicin* (if *penicillin* allergic) *Vancomycin* + *gentamicin* + *rifampin* if early prosthetic valve endocarditis (<1 year) or healthcare-associated IE	
*S. aureus*	**Methicillin-sensitive**	**Methicillin-resistant**
	Native: *oxacillin* or *cefazolin*/*cefotaxime* 4–6 w PVE: *oxacillin* 6 w + *rifampin* 6 w + *gentamicin* 2 w OR *cefazolin*/*cefotaxime* 6 w + *rifampin* 6 w + *gentamicin* 2 w if *penicillin* anaphylaxis OR *vancomycin* 6 w + *rifampin* 6 w + *gentamicin* 2 w if *penicillin* anaphylaxis	*vancomycin* or *daptomycin* 4–6 w
*Streptococcus*	***Penicillin*-sensitive**	***Penicillin*-resistant**
	Native: *penicillin* G or *ceftriaxone* or *amoxicillin* 4 w OR *vancomycin* 4 w (*penicillin* allergic) PVE *: *penicillin* G or *ceftriaxone* or *amoxicillin* 6 w OR *vancomycin* 6 w (*penicillin* allergic)	Native: *penicillin* G or *ceftriaxone* 4 w + *gentamicin* 2 w OR *vancomycin* +/− *gentamicin* 4 w (*penicillin* allergic) PVE: *penicillin* G or *ceftriaxone* 6 w + *gentamicin* 2 w OR *vancomycin* 6 w + *gentamicin* 2 w (*penicillin* allergic)
Enterococcus	Non high-level aminoglycoside resistance *Ampicillin* + *ceftriaxone* 6 w *Vancomycin* + *gentamicin* 6 w (if *penicillin* allergic)	High-level aminoglycoside resistance *Ampicillin* + *ceftriaxone* 6 w *Amoxicillin* + *gentamicin* 4–6 w
HACEK group	*Ceftriaxone* OR *cefotaxime* OR *ampicillin-sulbactam* 4 w	Alternative regimen *Ampicillin*/*cephalosporin* + *aminoglycoside* 6 w
*Fungi Candida* spp., *Aspergillus* spp.	Cardiac surgery + antifungal agents *Amphotericin* B +/− *flucytosine* 6 w	Chronic suppressive therapy with oral fluconazole lifelong in patients who cannot undergo surgical resection

HACEK: *Hemophilus* species, *Aggregatibacter* species, *Cardiobacterium hominis*, *Eikenella corrodens*, and *Kingella* species. PVE: Prosthetic Valve Endocarditis.

**Table 7 jcm-11-03217-t007:** Indications for surgical intervention in pediatric infective endocarditis.

Indications for Surgery in Patients with Infective Endocarditis
Valve dysfunction resulting in symptoms of heart failure
Left-sided IE caused by *S. aureus*
*Fungal* or highly resistant microorganisms
Complications: heart block, annular or aortic abscess, pseudoaneurism or fistulae
Persistent infection (persistent bacteremia, fever > 5–7 days despite appropriate antimicrobial therapy)
Relapsing infection (recurrence of bacteremia after a complete antibiotic course)
Persistent vegetation and recurrent emboli despite appropriate antimicrobial therapy
Persistent fever

**Table 8 jcm-11-03217-t008:** Current indications for antibiotic prophylaxis to prevent infective endocarditis (IE).

Previous IE
Previous cardiac surgery and prosthetic material for cardiac valve or congenital heart defects repair
Prosthetic valves
Cyanotic congenital heart disease
Heart transplant with heart valve disease
Mechanical circulatory support

## Data Availability

Not applicable.

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
