# Peer review of "Pediatric Infective Endocarditis: A Literature Review"

_jcm, 2022, doi:10.3390/jcm11113217_

Round 1
Reviewer 1 Report
Dear authors,
First, I would like to express sincere gratitude to get an opportunity to review your manuscript.
The effort of the author is appreciated, as the topic is relevant and promising.
Congratulation for your review of the literature and for bringing light into the pediatric IE.
As a suggestion, the title of the manuscript should contain also one of the following “a narrative review” or “a literature review”.
Author Response
Reviewer 1
Dear authors,
First, I would like to express sincere gratitude to get an opportunity to review your manuscript.
The effort of the author is appreciated, as the topic is relevant and promising.
Congratulation for your review of the literature and for bringing light into the pediatric IE.
As a suggestion, the title of the manuscript should contain also one of the following “a narrative review” or “a literature review”.
Thank you for this comment. We have changed the title of the manuscript according to the Reviewer’s suggestion: “Pediatric infective endocarditis: a literature review.”

Reviewer 2 Report
The presented review on IE in children by Vincent, Luna and Martinez-Selles covers a topic of utmost interest to pediatric cardiologists and provides important information for the reader. In their manuscript they propose a well structured and comprehensiv summary on infective endocarditis in children. They outline clinical findings, diagnosis and treatment of IE along with insights on epidemiology and microbiological sources of infection.
Very nicely they support the reader with clinical pictures and echocardiography pictures to enhance the reading experience.
Aside from a minor spell check I would only make one minor comment. That is, to consider insertion of a table on the modified Duke criteria.
Author Response
Reviewer 2
The presented review on IE in children by Vicent, Luna and Martinez-Selles covers a topic of utmost interest to pediatric cardiologists and provides important information for the reader. In their manuscript they propose a well structured and comprehensiv summary on infective endocarditis in children. They outline clinical findings, diagnosis and treatment of IE along with insights on epidemiology and microbiological sources of infection.
Very nicely they support the reader with clinical pictures and echocardiography pictures to enhance the reading experience.
Thank you for this comment.
Aside from a minor spell check I would only make one minor comment. That is, to consider insertion of a table on the modified Duke criteria.
As the Reviewer suggested, we have included a table on the modified Duke criteria.
Table 3. Modified Duke criteria for diagnosis of infective endocarditis (IE).
Infective endocarditis – modified Duke criteria* |
Major criteria |
Blood culture positive for typical microorganism (i.e. Staphylococcus aureus, Enterococcus, streptococci viridans) |
Echocardiography with valvular vegetation |
Minor criteria |
Predisposing cardiac condition or injection drug use |
Temperature >38º C |
Embolic phenomena |
Immunologic phenomena (glomerulonephritis, Osler’s nodes, Roth’s spots, and rheumatoid factor) |
Positive blood culture not meeting above criteria |
* Definite IE: 2 major OR 1 major + 3 minor criteria. Possible IE: 1 major + 1 minor OR 3 minor criteria

Reviewer 3 Report
In this review about endocarditis the authors consider the problems of infective endocarditis but only in the pediatric population. Some interesting points are properly addressed as in the clinical presentation and in microbiology sections of the paper. However a certain criticism has to be manifested.
1) the authors should declare how they search the literature (in particular the keywords, the span of time chosen, the chosen data base(s) ) and keep track of the search items you use (so that your search can be replicated); the field of infective endocarditis has got a vast coverage in PubMed (1,809 results by inserting as keywords pediatrics and endocarditis); so the specifications regarding how the literature were searched are of aid in understanding how robust is the coverage of the review paper.
2) One point that deserves improvement is that regarding diagnostic techniques in assessing infective endocarditis (IE); the authors mentioned as ultrasounds based approaches only transthoracic and transesophageal echocardiography; they underscore the limitations of these approaches for right sided form of endocarditis especially in patients with conduits, prosthesis, complex congenital disease, leads, catheters; then they indicate tomographic radiology such as CT and PET as possible further diagnostic steps in these patients in which ultrasound is not diagnostic although they underscore the limitations of these radiologic approaches since the big radiation exposure that is deadly dangerous especially in pediatric population. But they fail to consider the intracardiac echocardiography (ICE) approach, much more practical that transesophageal in pediatric population and with no radiation exposure differently to PET and CT; and this intracardiac ultrasound approach has been recently proved very effective in visualizing leads endocarditis mass (both infective and post-lead extraction) [1,2]; so the use of ICE in endocarditis must be properly covered and the paper must be improved in this respect.
3) That part concerning the pathogenesis of IE is not satisfying and should be improved. The authors say that one major risk factor is “Predisposing heart disease generating turbulent blood flow that produces a host immune response”. That is not clear. The jet lesions in congenital heart disease produce endocardial damage that causes a first thrombotic and fibrotic reaction aimed at repairing the damage (phase 1 of endocarditis); such jet induced thrombosis with platelet deposition and fibrin is the first of 4 events bringing about infective endocarditis by creating a minimal non-bacterial thrombotic vegetation (NBTV). So if there is bacteremia (2nd event), adherence of micro-organisms to the NBTV takes place, with subsequent deposition of more fibrin and platelets that cover the infective agents (3rd event) favoring their multiplication (4th event) [3-5]. This part so should be improved and more clearly reported.
4) In my view it should be further emphasized the difference and peculiarity of endocarditis, if any, in this young population with respect the adult population. So the first statement in each paragraph should be if peculiarity exists or does not in the pediatric population; in the risk factors coverage, for example, it is not clear which risk factors are really peculiar in the young population.
REFERENCES
1. Caiati C, Pollice P, Lepera ME, Favale S. Pacemaker Lead Endocarditis Investigated with Intracardiac Echocardiography: Factors Modulating the Size of Vegetations and Larger Vegetation Embolic Risk during Lead Extraction. Antibiotics (Basel, Switzerland). 2019;8(4).
2. Caiati C, Luzzi G, Pollice P, Favale S, Lepera ME. A Novel Clinical Perspective on New Masses after Lead Extraction (Ghosts) by Means of Intracardiac Echocardiography. Journal of clinical medicine. 2020;9(8).
3. Bayliss R, Clarke C, Oakley CM, Somerville W, Whitfield AG, Young SE. The microbiology and pathogenesis of infective endocarditis. Br Heart J. 1983;50(6):513-9.
4. Freedman LR. The pathogenesis of infective endocarditis. J Antimicrob Chemother. 1987;20 Suppl A:1-6.
5. Freedman LR, Valone J, Jr. Experimental infective endocarditis. Prog Cardiovasc Dis. 1979;22(3):169-80.
Author Response
Reviewer 3
In this review about endocarditis the authors consider the problems of infective endocarditis but only in the pediatric population. Some interesting points are properly addressed as in the clinical presentation and in microbiology sections of the paper. However a certain criticism has to be manifested.
We would like to thank Reviewer 3 for the comments that have helped us improve our manuscript.
- the authors should declare how they search the literature (in particular the keywords, the span of time chosen, the chosen data base(s) ) and keep track of the search items you use (so that your search can be replicated); the field of infective endocarditis has got a vast coverage in PubMed (1,809 results by inserting as keywords pediatrics and endocarditis); so the specifications regarding how the literature were searched are of aid in understanding how robust is the coverage of the review paper.
We have carried out a narrative review, including the most recently published evidence on infective endocarditis in general, and its application to the pediatric population, as well as studies specifically carried out in pediatric patients. We used Pubmed, Scopus, WOS and Cochrane. Keywords: “endocarditis”, “pediatric”, “children”, “congenital heart disease”. We also used the references we found in our first search. We have made the clarification in the first part of the manuscript, as follows: “This is a narrative review, including the most recent evidence and scientific advances in pediatric IE”.
In accordance with the reviewer 1 request, we have modified the title of the manuscript for further specification: “Pediatric infective endocarditis: a literature review.”
- One point that deserves improvement is that regarding diagnostic techniques in assessing infective endocarditis (IE); the authors mentioned as ultrasounds based approaches only transthoracic and transesophageal echocardiography; they underscore the limitations of these approaches for right sided form of endocarditis especially in patients with conduits, prosthesis, complex congenital disease, leads, catheters; then they indicate tomographic radiology such as CT and PET as possible further diagnostic steps in these patients in which ultrasound is not diagnostic although they underscore the limitations of these radiologic approaches since the big radiation exposure that is deadly dangerous especially in pediatric population. But they fail to consider the intracardiac echocardiography (ICE) approach, much more practical that transesophageal in pediatric population and with no radiation exposure differently to PET and CT; and this intracardiac ultrasound approach has been recently proved very effective in visualizing leads endocarditis mass (both infective and post-lead extraction) [1,2]; so the use of ICE in endocarditis must be properly covered and the paper must be improved in this respect.
We have included information related to intracardiac echocardiography and we have added the suggested references. "Intracardiac echocardiography is a promising tool in the diagnosis of IE. As we have previously mentioned, conventional transthoracic and transesophageal echocardiography may have poor diagnostic performance in children with cardiac devices or with previous corrective surgeries. In these cases, intracardiac echocardiography, which is a very versatile technique, can overcome these limitations."
- That part concerning the pathogenesis of IE is not satisfying and should be improved. The authors say that one major risk factor is “Predisposing heart disease generating turbulent blood flow that produces a host immune response”. That is not clear. The jet lesions in congenital heart disease produce endocardial damage that causes a first thrombotic and fibrotic reaction aimed at repairing the damage (phase 1 of endocarditis); such jet induced thrombosis with platelet deposition and fibrin is the first of 4 events bringing about infective endocarditis by creating a minimal non-bacterial thrombotic vegetation (NBTV). So if there is bacteremia (2nd event), adherence of micro-organisms to the NBTV takes place, with subsequent deposition of more fibrin and platelets that cover the infective agents (3rd event) favoring their multiplication (4th event) [3-5]. This part so should be improved and more clearly reported.
We have rewritten the pathogenesis paragraph to provide a more precise explanation: "The turbulent blood flow caused by structural lesions in congenital heart disease might produce endocardial damage that could cause a first thrombotic and fibrotic reaction aimed at repairing tissue damage (first event). Such jet-induced thrombosis with platelet deposition and fibrin is the first of four concatenated events causing IE by creating a minimal non-bacterial thrombotic vegetation. In the case of bacteremia (second event), an adherence of micro-organisms to the non-bacterial thrombotic vegetation can take place, with subsequent deposition of more fibrin and platelets that cover the infective agents (third event) favoring their multiplication (fourth event) [29–31].”
- In my view it should be further emphasized the difference and peculiarity of endocarditis, if any, in this young population with respect the adult population. So the first statement in each paragraph should be if peculiarity exists or does not in the pediatric population; in the risk factors coverage, for example, it is not clear which risk factors are really peculiar in the young population.
We have rewritten the first statement in each paragraph in order to point out the peculiarities of infective endocarditis in children:
“Globally, the frequency of IE in children is lower than in adults, but in recent decades an increase has been observed due to the rise in risk factors, such as congenital heart disease or intravascular devices (central catheters, pacing leads...). The predisposing factors are similar to those of the adult population, however the vulnerability to some of these factors may be greater in the pediatric population. In fact, younger age is a predisposing factor for IE in cases of bacteremia, and younger patients are at greater risk of worse outcomes [6].”
“About 50 – 70 % of pediatric IE is seen in patients with congenital heart disease [7–9], which is the main predisposing condition. Conversely, in adult patients, the most common heart diseases predisposing to IE include degenerative valve disease or valve replacement surgery with prosthesis implantation, which are significantly more common than in children [10]. (…) Congenital heart disease in adults with IE is uncommon but its prevalence is increasing, including complex congenital heart disease and patients with previous corrective surgeries, who have a high risk of IE, so an increase in the frequency of IE can be expected in this group [13,15].”
“In adults, degenerative valve disease and intravenous drug abuse are relevant risk factors for IE, although they are rare in children [16]. In adults without previous heart disease, chronic kidney disease on hemodialysis is a common risk factor for IE [17].”
“Oral health is an important factor in both children and adults. Episodes of transient bacteremia have been observed with dental procedures and with daily tooth brushing [25,26]. However, periodontal disease and gingivitis are more common in adults.”
“In a minority of cases, IE occurs in children with no known chronic conditions or congenital heart disease [27]. These previously healthy children are usually older compared to those with predisposing conditions [27]. Conversely, adults with no known risk factors tend to be younger [17].”
“IE is caused by an infection of the cardiac endothelium [28]. The pathophysiology is shared in adult and pediatric patients with IE.”
“The clinical presentation of IE in children is highly conditioned by age [20,28]. In adolescents, the symptoms and findings are quite similar to the ones seen in adult patients: fever, loss of appetite, and malaise as evidence of a systemic infectious process. (…) Immunological phenomena (Osler nodes, Janeway spots) are less common in children than in adults.”
“Causal pathogens vary according to the underlying conditions. In addition, there have been changes in IE epidemiology in recent decades. Globally, S. aureus remains the dominant causative pathogen both in children and adults [35]. Streptococci infections, primarily viridans group, are also very frequent [35]. Specifically, in the case of children, Staphylococcus species are the most common (especially in children without congenital heart disease), followed by Streptococcus species [1,10]. However, for children with underlying heart disease, viridans group Streptococcus is the most common cause [1,10] (figure 3). As in adults, IE from gram-negative organisms is rare in children [1,36,37].”
“IE is diagnosed based on the modified Duke criteria [41,42]. The diagnosis of IE is complex in children, because clinical manifestations are frequently nonspecific and may be confused with more common conditions. A high level of suspicion is required. The Duke criteria (table 3) combine clinical, echocardiographic, and microbiological findings. Duke criteria were primarily tested in adult patients and few studies that have evaluated the diagnostic performance of the current criteria for the diagnosis of pediatric IE [6,42,43]. The modified Duke classification is more sensitive in diagnosing IE in children [44] but, still, 12% failed to be classified as “definite” [42].”
“Complications can be classified as cardiac and extracardiac. Complications are more frequent in children without known heart disease, especially those <2 years of age [21]. Furthermore, complications of IE in children are less common in children than in adults [6].”
“The indications for surgery in the treatment of IE (table 7) in children are similar to those in adults.”
“Mortality in children with IE is still high, and the sequelae are also serious. Globally, mortality is around 1-5%, although it is higher in patients with IE complications [1,36,37,91,92]. In adults, mortality is even higher, about 10% [93].”
REFERENCES
We have included all the requested references, as the Reviewer’s suggested.
- Caiati C, Pollice P, Lepera ME, Favale S. Pacemaker Lead Endocarditis Investigated with Intracardiac Echocardiography: Factors Modulating the Size of Vegetations and Larger Vegetation Embolic Risk during Lead Extraction. Antibiotics (Basel, Switzerland). 2019;8(4).
- Caiati C, Luzzi G, Pollice P, Favale S, Lepera ME. A Novel Clinical Perspective on New Masses after Lead Extraction (Ghosts) by Means of Intracardiac Echocardiography. Journal of clinical medicine. 2020;9(8).
- Bayliss R, Clarke C, Oakley CM, Somerville W, Whitfield AG, Young SE. The microbiology and pathogenesis of infective endocarditis. Br Heart J. 1983;50(6):513-9.
- Freedman LR. The pathogenesis of infective endocarditis. J Antimicrob Chemother. 1987;20 Suppl A:1-6.
- Freedman LR, Valone J, Jr. Experimental infective endocarditis. Prog Cardiovasc Dis. 1979;22(3):169-80.

Round 2
Reviewer 3 Report
The paper has been substantially improved.